# Identification and Characterization of Circular RNAs in *Brassica rapa* in Response to *Plasmodiophora brassicae*

**DOI:** 10.3390/ijms23105369

**Published:** 2022-05-11

**Authors:** Huishan Liu, Chinedu Charles Nwafor, Yinglan Piao, Xiaonan Li, Zongxiang Zhan, Zhongyun Piao

**Affiliations:** 1College of Horticulture, Shenyang Agricultural University, Shenyang 110866, China; liuhuishan1993@163.com (H.L.); 15909825891@163.com (Y.P.); gracesleen@163.com (X.L.); 2Center for Plant Science Innovation and Department of Biochemistry, University of Nebraska-Lincoln, Lincoln, NE 68588, USA; cnwafor2@unl.edu

**Keywords:** clubroot, disease resistance, *Brassica rapa*, circRNAs, *Plasmodiophora brassicae*

## Abstract

*Plasmodiophora brassicae* is a soil-borne pathogen that attacks the roots of cruciferous plants and causes clubroot disease. CircRNAs are noncoding RNAs, widely existing in plant and animal species. Although knowledge of circRNAs has been updated continuously and rapidly, information about circRNAs in the regulation of clubroot disease resistance is extremely limited in *Brassica rapa*. Here, Chinese cabbage (BJN 222) containing clubroot resistance genes (*CRa*) against *P*. *brassicae* Pb4 was susceptible to PbE. To investigate the mechanism of cicRNAs responsible for clubroot disease resistance in *B*. *rapa*, circRNA-seq was performed with roots of ‘BJN 222’ at 0, 8, and 23 days post-inoculated (dpi) with Pb4 and PbE. A total of 231 differentially expressed circRNAs were identified between the groups. Based on the differentially expressed circRNAs, the circRNA–miRNA–mRNA network was constructed using the target genes directly or indirectly related to plant resistance. Upregulated *novel_circ_000495* suppressed the expression of *miR5656-y*, leading to the upregulation of *Bra026508*, which might cause plant resistance. Our results provide new insights into clubroot resistance mechanisms and lay a foundation for further studies exploring complex gene regulation networks in *B*. *rapa*.

## 1. Introduction

Eukaryotic cells contain two main types of ribonucleic acids (RNAs): messenger RNAs (mRNAs) and noncoding RNAs (ncRNAs). mRNAs and ncRNAs are generated by genomic transcription, but ncRNAs, including microRNAs (miRNAs), circular RNAs (circRNAs), and long noncoding RNAs (lncRNAs), are not translated into proteins. Noncoding RNAs have emerged as important molecules for the transcriptional and post-transcriptional regulation of gene expression in plants [1,2].

CircRNAs have been identified as non-functional byproducts of genomic transcription in humans [3,4]; however, recent studies have suggested that they are widespread in eukaryotes and play an important role in the life activities of organisms [5]. Circular RNAs (circRNAs) are new endogenous noncoding RNAs produced from precursor messenger RNAs, characterized by a covalent bond connecting the 3′ and 5′ ends processed from back-splicing [6,7]. These circRNAs are more stable than linear RNAs and are resistant to RNase R because they lack 3′ or 5′ ends. CircRNAs can be classified into exon circRNAs, intronic circRNAs, exon–intron circRNAs, intergenic circRNAs, and antisense circRNAs based on their source. The proportions of these types of circRNAs vary among different species. In rice and *Arabidopsis*, exon circRNAs account for the greatest portion of circRNAs; intergenic circRNAs comprise the highest proportion of circRNAs found in kiwi, wheat, soybean, potato, and tomato [8,9,10,11,12]. Increasing evidence suggests that circRNAs widely exist in plant and animal species and can act as “microRNA (miRNA) sponges” and “competing endogenous RNAs” in regulating the function of microRNAs (miRNAs), splicing, and transcription, including the modification of gene expression [13,14,15,16,17,18]. The formation of circRNAs can also affect the splicing of their linear transcripts, thereby acting to regulate the transcription of their parental genes [19].

Studies have demonstrated that circRNAs exhibit specific expression patterns, and that the expression of circRNAs is induced by various stressors, such as abiotic and biotic stress, including pathogen invasion [20,21,22,23,24,25,26]. Zhou et al. (2019) showed that the expression of circRNAs changes considerably under abiotic stress [20]. Darbani et al. (2016) reported that circRNAs regulate genes that are involved in several aspects of cellular metabolism, such as hormonal signaling, intracellular protein sorting, carbohydrate metabolism, cell wall biogenesis, respiration, amino acid biosynthesis, transcription and translation, and protein ubiquitination [27]. In *Arabidopsis*, overexpressing the circular RNA (circGORK) hosted by the guard cell outward-rectifying K+ channel (GORK) gene resulted in a positive effect on drought tolerance [28].

Meanwhile, circRNAs show differential expression to biotic stress. For example, 584 circRNAs were found to be differentially expressed in kiwifruit after canker pathogen infection [29]. Additionally, a recent study confirmed that the differential expression of circRNAs might play a role during the potato–*Pectobacterium carotovorum* subsp. *brasiliense* (*Pcb*) interaction [30]. A total of 280 differentially expressed circRNAs were identified in cotton infected with Verticillium wilt. The number of differentially expressed circRNAs in the susceptible line was approximately twice that in the disease-resistant line, and the differential expression levels of the circRNAs were generally higher in the disease-resistant line than in the susceptible line [31]. Wang et al. (2018) identified 32 and 83 differentially expressed circRNAs between tomato leaves infected with the yellow leaf curl virus (*YLCV*) and the control, respectively, and the expression level of circRNAs after infection with the virus was lower than that of the control [32]. Together, these results indicate that circRNAs might have essential functions in plant growth and development, as well as in disease response, which supports their role as novel interactors in regulating gene expression in plants.

Clubroot, caused by the soil-borne obligate intracellular parasite *Plasmodiophora brassicae*, is a severe and worldwide disease of the *Brassicaceae* [33]. The pathogen is known to possess a complex lifecycle and can remain in the soil for over 20 years as a resting spore [34]. The severe root damage caused by *P*. *brassicae* dramatically restricts water and nutrient transport from the roots, resulting in stunted plants and tremendous yield loss. Despite the existence of comprehensive management approaches, it is crucial to explore the molecular mechanism of the occurrence of clubroot. Some cruciferous plants have acquired a certain degree of defense mechanisms during evolution, including resistance genes, secondary metabolites, cell wall modifications, and plant hormone signaling pathways, in response to *P*. *brassicae* [35]. In the evolution of plants and pathogens, plant cytochrome P450 can catalyze a variety of primary and secondary metabolic reactions in plants, increase the resistance of plants to biotic and abiotic stimuli, and act as stress signals, activating disease resistance in plants. Cell lignification is a method of cell wall modification in plants. CYP941 of the P450 family catalyzes C18, which enhances the ability of plants to resist pathogen infection [36]. Isoflavone synthase is a precursor for the synthesis of phytochemicals in legumes, which belong to the P450 family. Evidence shows that the catalytic product of CYP740 plays an important role in potato resistance to blight [37]. Therefore, the P450 family is widely involved in the process of plant disease resistance. However, reports about the relationship between ncRNAs and the P450 family are limited. Recently, lncRNAs have been detected in resistant plant roots, which are known to harbor a quantitative trait locus conferring resistance to different pathotypes [38]. Verma et al. (2014) identified putative miRNA-regulated genes with roles during clubroot disease initiation and development, which required further experimental validation [39]. However, compared with mRNA, miRNA, and lncRNA, there is still a significant gap in our understanding of circRNAs, especially the role of circRNAs in plant clubroot resistance.

Chinese cabbage ‘BJN 222’ harboring the *CRa* resistant gene is resistant to *P*. *brassicae* pathotype 4 but susceptible to pathotype E, which was confirmed in the inoculation experiment. To identify and characterize the circRNAs involved in clubroot disease conditioning in *Brassica rapa*, we inoculated Chinese cabbage (BJN 222) with two *P*. *brassicae* pathotypes and identified and analyzed circRNAs from three stages: control check (ck), an early time of disease initiation (8 dpi), and a later time that the pathogen colonized the roots (as evident by visible gall formation) (23 dpi). We detected more than 1000 circRNAs, among which over 200 were differentially expressed, suggesting that circRNAs were effective indicators of plant resistance. In addition, differentially expressed circRNAs were identified, and their parental functional annotations were analyzed. Moreover, circRNA-originating target miRNA predictions were made to predict the function of circRNAs in *B*. *rapa*. These results lay a foundation for further research on the function of circRNAs responsible for pathogen infection.

## 2. Results

### 2.1. Identification and Validation of CircRNAs

A total of 15 RNA-seq libraries (Table 1) were constructed and sequenced in this study. A total of 1636 novel circRNAs were detected from our circRNA-seq data and named from *novel_circ_000001* to *novel_circ_001636* after BLAST searches against the circBase database [35,36]. The circRNAs detected in our study are shown in Appendix A.

According to genome origination, the 1636 circRNAs were classified into six types, namely annot_exons, one_exon, intronic, exon_intron, intergenic, and antisense, containing 100, 342, 13, 408, 201, and 572 circRNAs, respectively (Figure 1A). The number of circRNAs encoded by the antisense and intronic regions accounted for the most and the least circRNAs (34.9% and 0.7%), respectively. The annot_exons, one_exon, and exon_intron-type circRNAs containing the exon sequence accounted for most of the 1636 circRNAs (approximately 51.96%). These results are consistent with studies in other species, such as *Arabidopsis thaliana* and *Oryza sativa*, whose circRNAs originated from exons of a single protein-coding gene, accounting for 50.5% and 85.7%, respectively [37]. The distribution of these circRNAs on chromosomes of *B*. *rapa* ranged from 81 to 255 (Figure 1B). For example, 255 circRNAs from chromosome A06 accounted for the most (15.59%), followed by chromosomes A01 and A09. The length distribution of these circRNAs ranged from 101 to 2000 bp. Furthermore, the largest number of circRNAs had lengths ranging from 201 to 300 bp (Figure 1C).

To validate the circular RNAs detected from RNA sequencing, several circRNAs were randomly selected for polymerase chain reaction amplification. A pair of convergent primers was designed to amplify the linear DNA fragments when cDNA or genomic DNA was used as a template in the PCR reaction. Unlike convergent primers, the reverse primers of the divergent primers were located upstream of the forward primers (Figure 2A). The amplification products were not detected in the genomic DNA samples with divergent primers (Figure 2B). In contrast, there were no amplification products in the cDNA digested by RnaseR with convergent primers. After confirmation by PCR, sequencing analyses were used to confirm the junction sites of the PCR products (Figure 2C). We randomly selected 20 circRNAs at different stages for qRT-PCR to validate the expression levels. The qRT-PCR results were mainly (17/20) consistent with the circRNA-seq data, indicating the high reliability of the RNA profiles (Appendix A).

### 2.2. CircRNA Analysis in Response to P. brassicae

To explore the expression pattern of circRNA in response to the different pathotypes of *P*. *brassicae* in *B*. *rapa*, the circRNA expression profiles of all samples were compared by cluster analysis. The expression pattern of circRNA was divided into two clusters based on infection time (Appendix A). At the later stage of inoculation, the circRNA response to the two pathotypes of *P*. *brassicae* showed obvious differences. In the same cluster, there was a significant difference between non-inoculated samples and 8-day post-inoculation samples, regardless of the pathotype, suggesting that the circRNA expression pattern was affected by the inoculation time and pathotype of *P*. *brassicae* (Appendix A). A total of 231 significantly differentially expressed circRNAs (DE circRNAs) were identified between the 8 comparisons. There was no significant difference in the number of up- or downregulated circRNAs between the samples inoculated with PbE or Pb4 at 8 dpi (24 versus 27). As the inoculation time increased, the number of DE circRNAs increased to 118 in the samples inoculated with PbE; 76 circRNAs were upregulated, and 42 circRNAs were downregulated (Figure 3A). However, the number of DE circRNAs was only 64 at 23 dpi when inoculated with Pb4. The difference in the number of DE circRNAs implied that *B*. *rapa* was sensitive to infection by the virulent *P*. *brassicae* pathotype, even with the *CRa* locus. Overall, the number of DE circRNAs infected with PbE was greater than that of circRNAs infected with Pb4. These results highlight the distinct responses of ‘BJN 222’ containing *CRa* to pathogens exhibiting varying degrees of virulence.

Figure 3B shows the results of the dataset overlap. Compared with ck, there were 18 and 15 DE circRNAs at 8 dpi after inoculation with Pb4 and PbE, respectively. Furthermore, we identified 22 and 76 DE circRNAs at 23 dpi in samples inoculated with Pb4 and PbE, respectively. In addition, 25 circRNAs were co-expressed after inoculation with Pb4 or PbE at a comparison of 8 dpi and 23 dpi. Only 1 circRNA was found to be continuously expressed after inoculation with PbE at the two time points. Regardless of which pathotype was used for inoculation, the number of DE circRNAs at 23 dpi was much larger than that at 8 dpi, suggesting that *B*. *rapa* responded differently over time. Specifically expressed circRNAs in plants inoculated with PbE were 4-fold higher than those in Pb4 at 23 dpi, suggesting that *B*. *rapa* responded differently to different *P*. *brassicae* pathotypes. Together, these results suggest that circRNAs participate in the transcriptome-wide molecular landscapes of *B*. *rapa* responses to *P*. *brassicae* stress.

### 2.3. Identification of CircRNA Parental Genes

The annotated genes producing circRNAs are referred to as the parental genes of the circRNAs, while the circRNAs without parental genes were described as ‘NA’. Here, 1435 of the identified 1636 circRNAs originated from 1004 parental genes, and 201 intergenic-type circRNAs originated from the fragment between two genes without parental genes (Appendix A). A total of 826 circRNAs had only one parental gene, and 609 circRNAs originated from more than one parental gene. The parental genes produced different circRNAs due to the alternative splicing pattern in *B*. *rapa*, which is consistent with previous studies suggesting that circRNAs possess an alternative splicing pattern, making them a valuable resource for understanding the complexity of circRNA biogenesis and their potential functions [40,41].

### 2.4. CircRNA Parental Gene Function Analysis

CircRNAs play a significant role in transcriptional control by the cis-regulation of their host genes [42]. To gain insight into the potential circRNA-mediated mechanism of *B*. *rapa* response to infection by *P*. *brassicae*, KEGG pathway enrichment analysis was employed to analyze the DE circRNA parental genes (Figure 4A). The comparison between the avirulent Pb4 and the control group at 8 dpi showed that the DEGs were enriched in “Sulfur metabolism” (ko00920), “Microbial metabolism in diverse environments” (ko01120), and “Citrate cycle (TCA cycle)” (ko00020) (*p* < 0.05). However, no significant pathway enrichment was detected at 23 dpi between the two groups. We also compared the virulent PbE and control group at 8 dpi, and one significant pathway enrichment was detected i.e., “Sulfur metabolism” (ko00920) (*p* < 0.05), while the pathway “Vitamin B6 metabolism” (ko00750) was enriched at 23 dpi. The pathway “Phenylalanine, tyrosine and tryptophan biosynthesis” (ko00400) was significantly enriched in the comparison of PbE vs. Pb4 at 8 and 23 dpi. Tryptophan biosynthesis is involved in SA and camalexin biosynthesis. The plant hormone salicylic acid (SA) plays a critical role in defense against biotrophic pathogens, such as *Plasmodiophora brassicae* [43], and camalexin is a sulfur-containing tryptophan-derived secondary metabolite reported to play defensive functions against several pathogens in *Arabidopsis* [44,45].

Gene Ontology (GO) analysis was performed on the parental genes to understand the biological function of the DE circRNAs (Figure 4B). The parental genes were classified into three GO categories: biological process, molecular function, and cellular component. The top five subcategories in the biological processes class were “cellular process”, “single-organism process”, “metabolic process”, “response to stimulus” and “localization”, implying that the parental genes of circRNAs may function in response to the infection of *P*. *brassicae*. The most highly represented molecular function categories were “catalytic activity” and “binding.” The top 3 subcategories in the cellular component class were “cell”, “cell part” and “organelle”. Similarly, in the KEGG pathway enrichment analysis, “metabolic process” and “response to stimulus” were overrepresented in all comparison groups, and the number of DEGs was significantly higher at 23 dpi than at 8 dpi.

Together, many parental genes responded to *Rhizobium* stress. Additionally, the number of DE circRNAs varied considerably over time, but most pathways of enrichment were associated with metabolic processes, which may be due to plant life activities rather than stimulation by *P*. *brassicae*. Therefore, we focused on the circRNAs and related genes of significantly enriched pathways at the early stage of infection with *P*. *brassicae*. The parental genes and their annotations in each comparison are shown in Table 2.

### 2.5. CircRNAs Acting as MiRNA Sponges

To determine whether circRNAs in *B*. *rapa* could affect post-transcriptional regulation by binding to miRNAs and preventing them from regulating their target mRNAs, we identified miRNA target sites of circRNAs in *B*. *rapa* and found that 257 of 1636 (15.7%) circRNAs had putative miRNA-binding sites (Appendix A). Of these 257 circRNAs, 116 had more than one different miRNA-binding site, and the greatest number of miRNA-binding sites (39) was found in *novel_circ_000502*. *miR5021-x* and *miR2275-x* accounted for most of the circRNAs (55 and 38, respectively). Similarly, *novel_circ_000**495* and *novel_circ_001061* were differentially expressed between Pb4- and PbE-inoculated plants at 8 dpi, with 3 and 2 miRNA-binding sites, respectively (Figure 5). These results indicate that circRNAs in *B*. *rapa* have many potential miRNA-binding sites and probably affect the expression of disease resistance genes through miRNA.

### 2.6. Construction of the CircRNA–MiRNA–MRNA Network

To better understand the gene regulatory network during the infection of *Brassica rapa*, DE circRNAs and their target genes in the PbE-8d-vs.-Pb4-8d group were selected for circRNA–miRNA–mRNA network analysis with Cytoscape software (http://www.cytoscape.org/ (accessed on 19 May 2021)) (Appendix A). By predicting the miRNAs and their target genes of circRNAs, many miRNA-targeted mRNAs were annotated as stress-associated. A total of 20 target genes were annotated with “receptor-like protein kinase”, “zinc finger protein”, “LRR-repeat protein” and “hormone and P450” functions (Appendix A). The relative expression levels of these candidate target genes showed diverse expression patterns (Figure 6). Here, 4 target genes, *Bra011339* (zinc finger protein), *Bra013568* (zinc finger protein), *Bra026508* (hormone-related), and *Bra036269* (LRR-repeat protein), were consistent with the expression pattern of *novel_circ_000495* at 8 dpi (Figure 7). *Bra011339* is the homolog of *At4g32340*, encoding a tetratricopeptide repeat (TPR)-like superfamily protein, and it is widely involved in biological processes, such as cell cycle regulation, protein transport, mitosis, steroid receptor function, RNA splicing, transcriptional repression, protein degradation, and stress. *Bra013568* is annotated with the zinc finger *MYND* domain protein. However, related studies on plants are rare. *Bra036269* is the homolog of *At4g02410*, which is annotated with *LECRK43*. Studies have shown that overexpressing plants display increased resistance to infection by *Botrytis cinerea* compared to wild-type *Arabidopsis thaliana* [46]. However, only *Bra026508* (homolog of *At4g15330*, which annotated with *CYP705A5*) showed significant differential expression between plants inoculated with Pb4 and PbE, and Pearson’s correlation analysis suggested a positive correlation between *Bra026508* and *novel_circ_000495* under *P*. *brassicae* infection (r = 0.85), indicating that expression of the target genes could be enhanced in response to *P*. *brassicae* infection. Therefore, the circRNA *novel_circ_000495* probably plays an important role in clubroot resistance through post-transcriptional control of its target genes, which are involved in biotic stress. Furthermore, further research is needed to confirm the function of *Bra026508* in the *P*. *brassicae* resistance mechanism, and other disease-related genes are also worth exploring.

## 3. Discussion

### 3.1. CircRNA Sequencing Explains the Mechanisms of Clubroot Resistance in B. rapa

Although recent studies have made some progress in ncRNAs, the mechanism of Chinese cabbage clubroot resistance remains poorly understood. Sequencing of noncoding RNAs may provide deep insights into plant disease resistance mechanisms. Here, the *B*. *rapa* disease-resistant variety ‘BJN 222’, harboring the *CRa* gene, was inoculated with avirulent Pb4 or virulent PbE *P*. *brassicae*; transcriptome sequencing revealed many differentially expressed circRNAs at 0, 8, and 23 dpi. A total of 1636 novel circRNAs were identified during the *P*. *brassicae–B*. *rapa* interaction and the contributions of the five groups of *B*. *rapa* to the generated circRNAs were approximately the same, indicating that circRNAs are universally present in *B*. *rapa*. Our results suggest that significant changes occurred in the transcriptome of *B*. *rapa* during infection with the two pathotypes, especially in the expression levels of many genes detected in different disease conditions. For example, the number of differentially expressed circRNAs at 23 dpi was significantly higher than that at 8 dpi, which is consistent with Fu et al. Additionally, the number of differentially expressed circRNAs in PbE at 23 dpi was significantly higher than that in Pb4 (118 to 64), indicating that Chinese cabbage was sensitive to virulent *P*. *brassicae* infection. How these differentially expressed circRNAs are related to the resistance of *P*. *brassicae* infection still needs to be verified in subsequent experiments.

Studies in rice have shown that multiple circRNAs can originate from a single gene [8] linked to alternative splicing [46,47,48,49]. In this study, multiple circRNAs from one parental gene also occurred, indicating that alternative back-splicing occurred in *B*. *rapa*. Gao et al. (2016) suggested that alternative splicing events in circRNAs are not always consistent with the corresponding mRNAs [41]; therefore, the biogenesis, regulation, and function of alternatively spliced circRNAs in plants are worthy of further study [50,51,52]. Although circRNAs may not have the same function as parental genes, some studies have shown that their functions are closely related. For example, circRNAs can interact with RNA polymerase II and promote the transcription of their parental genes [46], reduce the expression of the source linear RNA [22], and promote exon skipping in their parental genes [53]. Therefore, most reports currently reflect the potential functions of circRNAs by studying the results of KEGG and GO function enrichment analyses of parental genes from which the circRNAs are derived [54,55]. In the present study, pathway enrichment revealed that there are many small molecular compounds, such as terpenoids, flavonoids, and alkaloids, which are related to disease resistance [56]. These include tropane, piperidine, and pyridine alkaloid biosynthesis (ko00960), ubiquinone and other terpenoid-quinone biosynthesis (ko00130), and the biosynthesis of amino acids (ko00400). These molecular compounds were found to be involved in the disease response, which might imply that changes in the cellular structure, nutrients, energy levels, and redox status occurred during fungal infection. Notably, sulfur metabolism (ko00920) was significantly enriched in the comparison between groups at 8 dpi, indicating that sulfur metabolism plays a role in the early stage of *P*. *brassicae* infection. A recent report confirmed that sulfur metabolism is involved in drought stress tolerance in broccoli [57]. Another study also found that primary metabolites and hormones participate in plant stress [58]. These genes were significantly enriched in the plant hormone signal transduction (ko04075) pathway in the group of ck-0d-vs.-8d-Pb4, indicating that hormone signaling plays a role in the early stage of plant inoculation. GO function analysis revealed the proportion of differentially expressed genes in the eight comparison groups. For example, molecular functions account for about 10.65%, biological processes account for about 54.74%, and cellular components account for about 34.61%. This is similar to the results of Chen et al. (2017) on the expression profile of soybean seeds at different developmental stages. Among the biological process GO terms, “response to stimulus” and “immune system process” were significantly enriched, and several of the genes that fell within these GO terms have been shown to be directly involved in disease resistance and stress response, implying that the parental genes of circRNAs participate in the response to *P*. *brassicae* [9].

### 3.2. CircRNAs Act as MiRNA Sponges Affecting the Function of Target Genes

CircRNAs can function as miRNA sponges, regulating the transcription of target genes [59,60]. For example, *ciRS-7* (circular RNA sponge for *miR-7*), which was identified in human and mouse brains, has been found to have more than 70 conventional miRNA target sites and acts as an miRNA sponge, regulating the function of *miR-7* [61,62]. The mode of action of miRNAs differs in plants and animals. Compared to plants, the range of target genes regulated by miRNAs is relatively wide in animals. Many miRNAs can bind to partially complementary target gene regions to inhibit their translation process. In plants, the binding of miRNAs and target genes requires strict complementary base pairing (usually 2–4 base pair mismatches), and the target mRNA is directly degraded after binding. This relationship is often one-to-one. Several microRNAs with known functions have been identified in plants. For example, mir156 has been reported to be involved in hormone-regulated pathways and in mediating responses to biotic and abiotic stress in *Arabidopsis* plants [63,64]. A *Brassica* miRNA, miR1885, targets both an immune receptor gene and a development-related gene for negative regulation through distinct modes of action [65]. Reports suggest that miRNA-mediated cleavage may contribute to the degradation and low abundance of circRNAs in plants [66]. Until now, most studies on circRNAs have reported them as competing endogenous RNA (ceRNA) molecules for predictive analysis in plants [67,68]. In this study, circRNAs differentially expressed under *Rhizobium* stress targeted the same miRNA, and multiple miRNAs targeted the same circRNA. Based on the results of KEGG and GO analysis, two significantly differentially expressed circRNAs were identified as having miRNA (*miR7121-x*, *miR5656-y*, *miR3699-y*) binding sites, implying that circRNAs might participate in clubroot disease response via miRNA pathways. Thus far, the role of miR5656 in clubroot disease has not been reported. Our study provides new insights into the clubroot disease resistance mechanism of Chinese cabbage.

### 3.3. CeRNA Networks Could Provide New Insights into the Regulatory Roles of NcRNAs during P. brassicae Infection

Recently, the regulation of circRNAs, miRNAs, and mRNAs has been confirmed in various diseases [69]. The first report on the identification of SSRs within lncRNAs responsive to infection demonstrates the potential use of lncRNAs in the breeding of Brassica crops [70]. However, the circRNA–miRNA–miRNA regulatory network has not been widely constructed in plants. To uncover the ceRNA network and the functions of circRNAs during *P*. *brassicae* infection in *B*. *rapa*, we constructed a putative circRNA–miRNA–mRNA network. From the network, we selected 4 candidate genes according to the expression level. These target genes are related to plant disease and might play an important role in clubroot resistance. Following the statistical analysis, we further analyzed *Bra026508*. *N**ovel_circ_000495* may act as an miRNA sponge by targeting *miR5656-y* to regulate the expression of *Bra026508*, whose function was cytochrome P450 705A5-like. Cytochrome P450 is a heme-containing multifunctional oxidase that binds CO in its reduced state and has an absorption peak at 450 nm. P450s perform two types of biosynthetic and metabolic detoxification functions in plants, some of which have essential roles in plant defense responses. CYP705A5 encodes an endomembrane system-expressed member of the CYP705 family of cytochrome P450 enzymes [71]. The interaction between CYP450 and clubroot disease is still rarely reported. Our study will give a new sight into the exploration of the relationship between the CYP450 family and ncRNAs. According to the study, the upregulation of the *novel_circ_000495* suppressed the expression of *miR5656-y*, leading to the upregulation of *Bra026508*. Upregulated *Bra026508* might affect plant resistance to different types of *P*. *brassicae* in *B*. *rapa*. Further work is needed to dissect the relationship between *novel_circ_000495*, *miR5656-y*, and *Bra026508* in vivo, and more studies should focus on the downstream effects of *Bra026508* to further clarify the mechanism of disease resistance.

## 4. Materials and Methods

### 4.1. Plant Materials and Inoculation with P. brassicae

Chinese cabbage ‘BJN 222’, harboring the *CRa* resistance gene, was resistant to pathotype 4 but susceptible to pathotype E of *P*. *brassicae*, which was confirmed in an inoculation experiment (Appendix A). To confirm successful infection, the susceptible line (BJN 3-2) was maintained in the climate chamber for 32 days after inoculation (Appendix A). To study the resistance mechanism of Chinese cabbage ‘BJN 222,’ the avirulent and virulent pathotypes of *P*. *brassicae*, Pb4, and PbE, respectively, were used to inoculate ‘BJN 222’.

The Pb4 and PbE of *P. brassicae* were kept in the College of Horticulture, Shenyang Agriculture University, Shenyang City, Liaoning Province, China. The Pb4 originated from Xinmin, Liaoning province, China (122° E, 41° N; China), and PbE originated from Shenyang, Liaoning province, China (123° E, 42° N; China). Both of them were maintained and propagated in the susceptible Chinese cabbage lines, and fresh galls were stored at −20 °C for further use. Preparation of the resting spores of *P. brassicae* was according to the method described by Pang et al. [72]. After the galls were ground in sterile distilled water with a homogenizer, the mixtures were filtered through 8 layers of cheesecloth. The resting spores were collected by centrifugation at 2500 g and quantified with a hemocytometer (Neubauer improved, Marienfeld, Germany). For inoculation, 1 ml of Pb4 or PbE spore suspension (1.0 × 10^7^ spores/mL) was applied to the stem base of 21-day-old Chinese cabbage plants. The inoculated and non-inoculated control plants were maintained in a climate-controlled room at 20–25 °C under a 16-h photoperiod. The soil was kept moist during the treatment period [73].

The roots of ‘BJN 222’ were collected at 0, 8, and 23 days post-inoculation (dpi) to analyze genes that were differently expressed between plants infected with Pb4 compared with PbE. The roots of 30 plants were sampled at each time point, and three independent biological replicates (10 plants for each replicate) were performed. The roots were washed with distilled water and immediately frozen in liquid nitrogen and stored at −80 °C. Ten plants from each replicate were pooled for further RNA extraction. To confirm the success of the infection, plants were maintained in a climate-controlled environment for 30 days after inoculation.

### 4.2. RNA and DNA Isolation, Library Preparation and Sequencing

According to the manufacturer’s protocol, the total RNA from all samples was extracted using TRIzol reagent (Invitrogen, Carlsbad, CA, USA). Total RNA was treated with DNase I (Takara Bio, Dalian, China) to remove contaminated genomic DNA. The RNA purity of each sample was determined using a NanoDrop-2000 spectrophotometer (Thermo Fisher Scientific, Wilmington, DE, USA). Samples with a 260/280 ratio of 1.9–2.1 and a 260/230 ratio ≥ 2.0 were considered of high quality and were used for experiments. RNA integrity was verified by 2.0% TAE agarose gel electrophoresis. For RNase R treatment, the total RNA was incubated for 15–30 min at 37 °C with 2 units RNase R per µg total RNA (Lucigen). RNA samples treated with or without RNase R were reverse transcribed with random primers using SuperScript II reverse transcriptase (Takara Bio, Dalian, China).

### 4.3. Mapping to the Reference Genome and Transcriptome Assembly

From both ends of the unmapped reads, 20-mers were extracted and aligned to the reference genome (Version 1.5, http://brassicadb.org/brad/datasets/pub/Genomes/Brassica_rapa/V1.0/V1.5 (accessed on 14 May 2019)) to determine the unique anchor positions within the splice site. Anchor reads aligned in the reverse orientation (head-to-tail) indicated circRNA splicing and were then subjected to find_circ [35] to identify circRNAs. The anchor alignments were then extended such that the complete read alignment and GU/AG splice sites flanked the breakpoints. A candidate circRNA was called if it was supported by at least two unique back-spliced reads in at least one sample.

To identify differentially expressed circRNAs across samples or groups, the edgeR package (http://www.rproject.org/ (accessed on 14 May 2019)) was used. The reads per million mapped (RPM) were used to evaluate the relative expression levels of the circRNAs. The expression levels of the circRNAs in the three replications were averaged as the result of one treatment. Significantly differentially expressed circRNAs were identified by the paired t-test, with *p* < 0.05 and |log_2_FC| (fold change/FC) > 1 between samples or groups.

### 4.4. CircRNA Detection and Functional Annotation

To evaluate the potential functions of the parental genes of circRNAs, the parental genes were annotated based on the GO (Gene Ontology) database and the KEGG (Kyoto Encyclopedia of Genes and Genomes) database. According to GO annotation, the genes were functionally categorized using BLAST2GO software [74]. The GO database comprises three ontologies: molecular function, cellular component, and biological process. Pathway enrichment analysis identified significantly enriched metabolic pathways or signal transduction pathways in parental genes compared with the whole genome background. Pathways meeting the condition of *p* < 0.05 were defined as significantly enriched pathways in parental genes.

### 4.5. CircRNA MiRNA-Binding Site Analysis

For circRNAs that have been annotated in circBase, the target relationship with miRNAs can be predicted using StarBase (v2.0). For novel circRNAs, postmatch software (v1.2) was used to predict the target genes for plant samples.

### 4.6. Integrated Analysis of CircRNAs–MiRNAs–MRNAs

To predict mRNAs interacting with circRNAs and miRNAs, miRTarBase (v6.1) was used to predict mRNAs targeted by the miRNA sponge. Cytoscape was used to visualize the resulting correlation of circRNAs–miRNAs–mRNAs.

### 4.7. CircRNA Validation and Quantitative Real-Time PCR

Primers (convergent and divergent) were designed using Primer5.0 software. The primers were further synthesized using Synbio Tech (Suzhou, China). Both genomic DNA (gDNA) and cDNA (RNase R+ or RNase R−) were used as templates for convergent and divergent primers. A real-time quantitative polymerase chain reaction (qRT-PCR) was performed to validate the expression of circRNAs. qPCR was carried out in a total volume of 20 µL, containing 10 µL of SYBR Premix ExTaq (2×), 0.5 µL of each primer (10 µM), 2 µL diluted cDNA, and 7 µL ddH2O (TIANGEN). Each sample was amplified with three technical replicates, and all PCR reactions were performed with the BIORAD CFX96 Real-Time System PCR. The qPT-PCR program was as follows: 95 °C for 15 min, followed by 45 cycles of 95 °C for 10 s, 60 °C for 25 s, and 72 °C for 30 s. The specificity of amplification was confirmed by melting curve analysis after 40 cycles. Analysis of gene expression was performed for all samples at 0, 8, and 23 dpi in ‘BJN 222’ inoculated with Pb4 and PbE. The 2^−ΔΔCT^ method was used to calculate the relative expression of circRNAs [75]. The primer sequences are listed in Appendix A.

## 5. Conclusions

In this study, the profile of circRNA expression was focused on during *P*. *brassicae* inoculation in *B*. *rapa.* Accordingly, we identified 1636 circRNAs, of which 231 showed significant differential expression. Functional characterization of circRNA host genes revealed that circRNAs were mainly related to the biosynthesis of terpenoids, flavonoids, and alkaloids. Based on the theory of ceRNA, we established disease-responsive ceRNA networks. These results indicated that circRNAs and their target genes were involved in modulation during the *P*. *brassicae* infection of *B*. *rapa*. The current study enhanced our understanding of the biological processes involved in *P*. *brassicae–B*. *rapa* interactions and may lead to new ideas for breeding disease-resistant phenotypes.

## Figures and Tables

**Figure 1 ijms-23-05369-f001:**
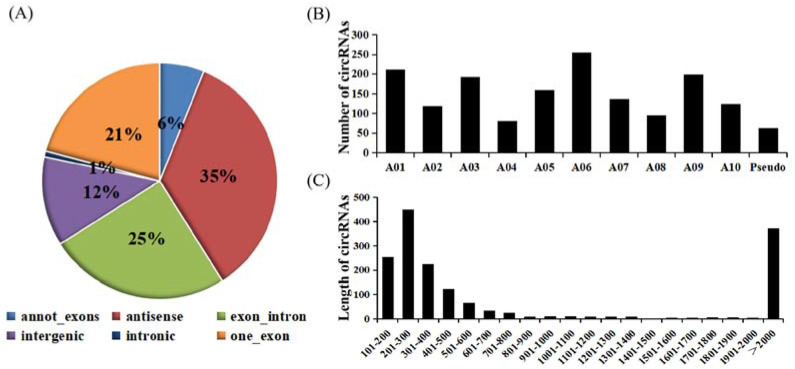
Characterization of circRNAs identified between control plants and plants infected with the two pathotypes of *P. brassicae* at 8 and 23 dpi. (**A**) Types of circRNAs. (**B**) Histogram of the distribution of circRNAs on the chromosome. (**C**) Distribution of the length of the circRNAs.

**Figure 2 ijms-23-05369-f002:**
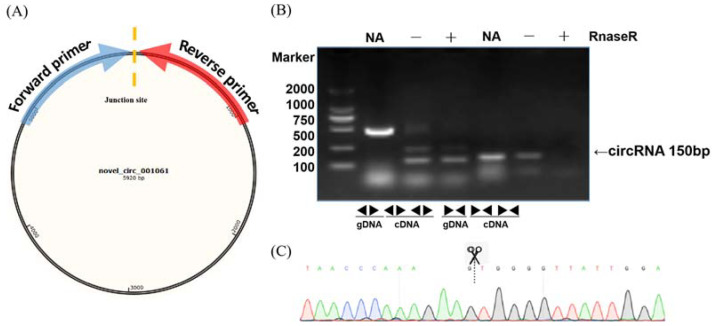
Validation examples of circRNA production. (**A**) Diagram of the circular RNA junction site. The arrow direction indicates the direction of the divergent primer design. (**B**) PCR reactions with divergent and convergent primers using different templates showing the production of *novel**_circ**_001061*. (**C**) Sequencing confirmation of the junction site of *novel**_circ**_001061*.

**Figure 3 ijms-23-05369-f003:**
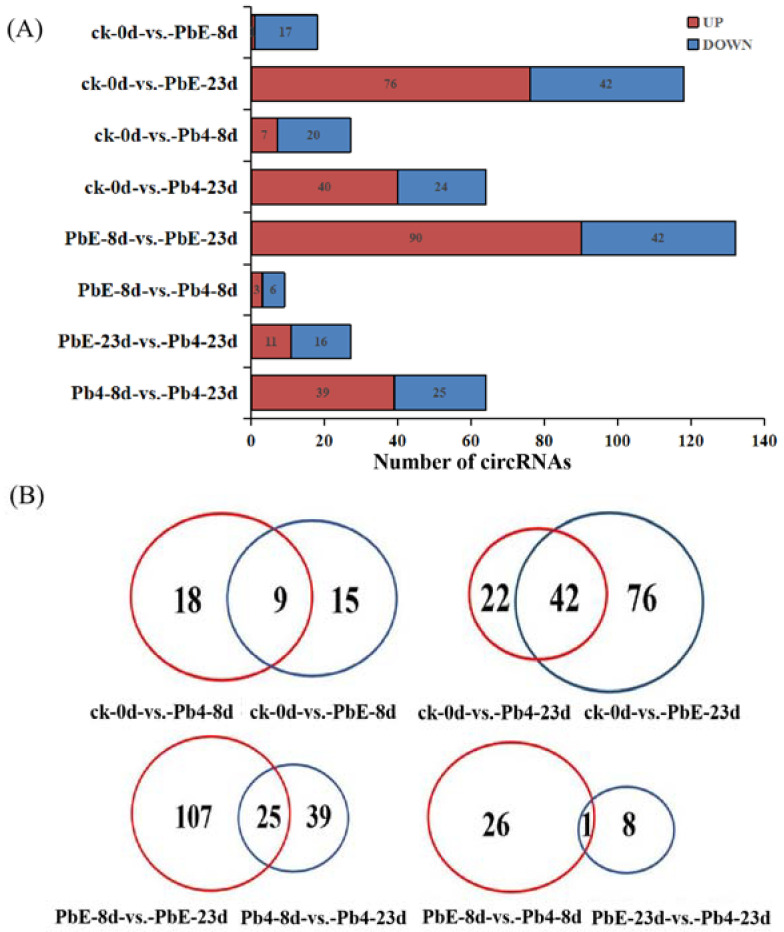
(**A**) Comparison of the number of circRNAs between control plants and plants infected with the two pathotypes of *P. brassicae* at 8 and 23 dpi. Blue and red represent up-and downregulated circRNAs, respectively. The results of 8 comparisons are shown. (**B**) Venn diagrams showing the overlaps of circRNAs differentially expressed in each comparison.

**Figure 4 ijms-23-05369-f004:**
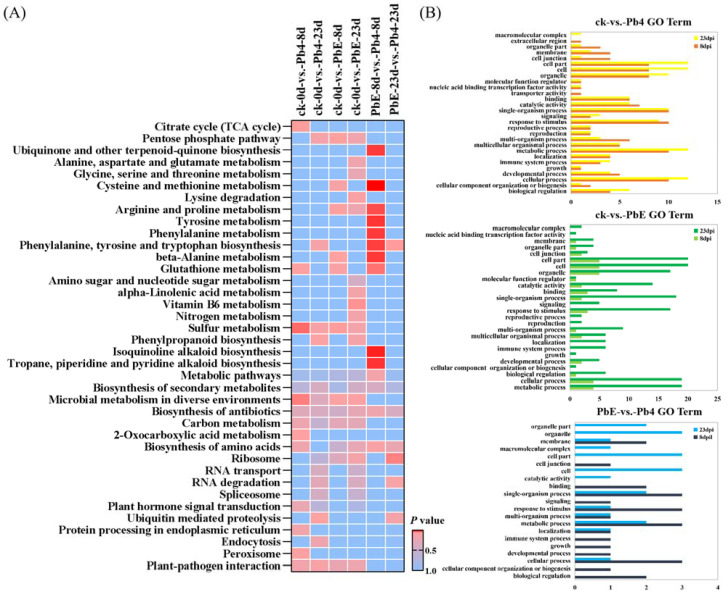
(**A**) KEGG pathway enrichment analysis of the differentially expressed parental genes in response to *P. brassicae* infection at 8 and 23 dpi. (**B**) Gene ontology (GO) classifications of the differentially expressed parental genes in response to *P. brassicae* infection at 8 and 23 dpi. GO classifications are listed on the left, whereas the gene numbers are shown on the right. GO classifications in the control vs. Pb4 comparison at 8 dpi and 23 dpi.

**Figure 5 ijms-23-05369-f005:**
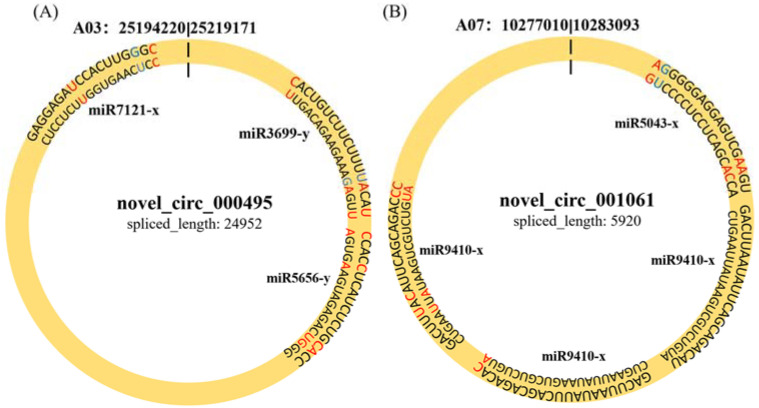
The character of sequence pairing structure between circRNA and their target miRNAs. (**A**) *novel_circ_000495*: *miR3699-y*, *miR5656-y*, *miR7121-x*. (**B**) *novel_001061*: *miR5043-x*, *miR9410-x*.

**Figure 6 ijms-23-05369-f006:**
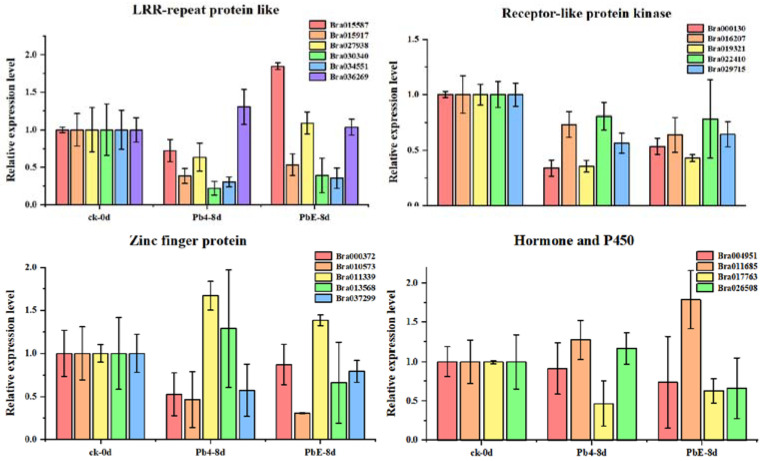
Expression of stress-associated genes at different stages.

**Figure 7 ijms-23-05369-f007:**
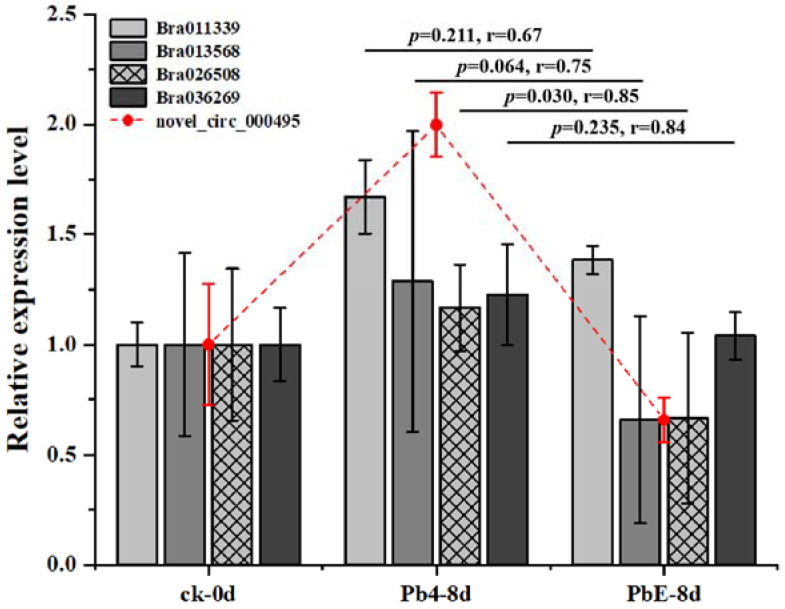
Relative expression level of target genes and circRNAs in roots.

**Table 1 ijms-23-05369-t001:** Profile of sample information.

Sample	Treatment	Replicate
ck-0d-1	control	biological replicate 1
ck-0d-2	control	biological replicate 2
ck-0d-3	control	biological replicate 3
Pb4-8d-1	inoculated with Pb4	biological replicate 1
Pb4-8d-2	inoculated with Pb4	biological replicate 2
Pb4-8d-3	inoculated with Pb4	biological replicate 3
PbE-8d-1	inoculated with PbE	biological replicate 1
PbE-8d-2	inoculated with PbE	biological replicate 2
PbE-8d-3	inoculated with PbE	biological replicate 3
Pb4-23d-1	inoculated with Pb4	biological replicate 1
Pb4-23d-2	inoculated with Pb4	biological replicate 2
Pb4-23d-3	inoculated with Pb4	biological replicate 3
PbE-23d-1	inoculated with PbE	biological replicate 1
PbE-23d-2	inoculated with PbE	biological replicate 2
PbE-23d-3	inoculated with PbE	biological replicate 3

**Table 2 ijms-23-05369-t002:** The circRNAs and their parental genes in each comparison.

Group	CircRNA ID	Source Gene ID	log_2_ (FC)	*p* Value	Chr	Strand
ck-0d-vs.-Pb4-8d	*novel_circ_000064*	*Bra013579*	4.246827502	0.004744418	A01	−
	*novel_circ_000079*	*Bra013579*	2.893516888	0.009588829	A01	−
	*novel_circ_000086*	*Bra013579*	−2.84065774	0.028360819	A01	+
	*novel_circ_000264*	*Bra039746*	−19.12803312	0.04701868	A02	+
ck-0d-vs.-PbE-8d	*novel_circ_000074*	*Bra013579*	19.51979677	0.011219066	A01	−
	*novel_circ_000086*	*Bra013579*	−3.947713015	0.012980797	A01	+
PbE-8d-vs.-Pb4-8d	*novel_circ_001061*	*Bra012389*	−4.222569626	0.038710775	A07	−
	*novel_circ_000495*	*Bra019293*	4.995712702	0.001038162	A03	−

## Data Availability

The raw sequence data reported in this paper have been deposited in the National Center for Biotechnology Information (NCBI) Sequence Read Archive (SRA) data under accession number PRJNA730971 (https://www.ncbi.nlm.nih.gov/sra/PRJNA730971 (accessed on 26 February 2022)).

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
