# Peer review of "Identification and Characterization of Circular RNAs in Brassica rapa in Response to Plasmodiophora brassicae"

_ijms, 2022, doi:10.3390/ijms23105369_

Round 1
Reviewer 1 Report
In this study, Authors identified and characterized the circRNAs involved in clubroot disease conditioning in Brassica rapa. They studied infected the Chinese cabbage (BJN 222) with two types of P. brassicae pathotype and identified and analyzed circRNAs from three stages (0 dpi, 8 dpi, 23 dpi). They detected more than one thousand circRNAs, among which over two hundred were differentially expressed, suggesting that circRNAs were effective indicators of plant resistance. The manuscript is well structured and well discussed. However, some points should be checked and corrected before its acceptance in this journal.
Therefore, according to my comments, I recommended the publication of the paper after major revision.
- The study's background should be clearly stated. Describe the introduction and review of the work.
- Please speculate on the results. The discussion must improve.
- Please provide in the conclusion section. The authors should add the significance of this research and its potential practical application.
- The MS English needs to be improved. The article's English must be carefully checked for grammatical errors.
Author Response
Dear reviewer:
On behalf of my co-authors, we thank you for giving us a chance to revise and improve the quality of our manuscript. We have read the reviewers’and your comments carefully and have made revision which using the “Track Changes” function in the paper.
We are very grateful to your comments for the manuscript. According with your advice, we have written a point-by-point response letter, please see the attachment.
In all, we found these comments are quite helpful. And special thanks to your comments again. I am looking forward to hearing from you.

Reviewer 2 Report
Manuscript entitled „Identification and Characterization of Circular RNAs in Brassica rapa in response to Plasmodiophora brassicae” presents the results of the study circRNA-seq concerning the roots of Chinese cabbage inoculated wit spores by avirulent and virulent pathotypes of P. brassicae.
I believe that the research is interesting and innovative and therefore deserves to be published.
I kindly ask you to complete the manuscript with information.
1.Where were avirulent and virulent pathotypes of P. brassicae obtained.
2.How are the strains stored? Are they in a collection in a specific collection?
3.How was the spore concentration determined?
4.What kind of spores were these? Zoospores?
5.Was infection of the plants confirmed? In what way?
6. Please include photographic material documenting the experiment in the supplementary files.
Author Response
Dear Reviewer,
Thank you for your letter and for the reviewers' comments concerning our manuscript entitled "Identification and Characterization of Circular RNAs in Brassica rapa in Response to Plasmodiophora brassicae". Those comments are all valuable and very helpfiul for revising and improving our paper, as well as the important guiding significance to our researches. We have studied comments carfully and we have written a point-by-point response letter, please see the attachment.
And special thanks to you and reviewers for your comments. Thank you for your consideration.I am looking forward to hearing from you.

Reviewer 3 Report
I started reviewing the manuscript. In the abstract line 19; I found the sentence "the circRNA-seq was performed roots of BJN 222 at 0 d, 8 d, and" which does not have any sense, but later on I found
line 33: "non codind"
Line 38 "hunams" and "resent"
This huge number of very obvious typos denotes that authors have not worked enough in the manuscript. This version is not suitable for a proper evaluation.
Author Response
Dear Reviewer,
On behalf of my co-authors, we thank you for giving us a chance to revise and improve the quality of our manuscript. We have read the reviewers’and your comments carefully and have made revision which using the “Track Changes” function in the paper.
According to comment, it is our negligence and we are sorry about these mistakes.
We have written a point-by-point response letter , please see the attachment.
Thank you for your consideration. I am looking forward to hearing from you.

Round 2
Reviewer 1 Report
Requested corrections were completed.
Author Response
Dear Reviewer:
We acknowledgment the time and expertise devoted to reviewing this manuscript. Those comments are all valuable and very helpful for revising and improving our paper. as well as the important guiding significance to our researches.
Once again, thank you very much for your comments and suggestions.
Yours sincerely,
Huishan Liu
Corresponding author:
Name: Zongxiang Zhan
Zhongyun Piao
E-mail: zhanzxiang@syau.edu.cn
zypiao@syau.edu.cn

Reviewer 3 Report
First of all, I want to confirm that this version is readable and that I appreciate the effort made by the authors to improve the manuscript. Even though, there are some points that should be addressed before being ready for acceptance.
- Line 14: Quotations (“,”) seem out of context.
- Line 68: “In Arabidopsis, overexpressing guard cell outward-rectifying K+ channel (circGORK) resulted in a positive effect on drought tolerance” It is not clear whether are speaking about the plant channel or to a circRNA. The mention to the plant channel seems strange. Please clarify.
- Figure 3: lettering too small.
- Figure 5: confusing. Which is the network? I only see too plasmids with a sequence, and again, lettering too small.
- Discussion: Authors have focused on the progress made in Arabidopsis, but seem to ignore some very recent advances in Brassicaceae. Authors report that they have found involvement of the citric acid cycle, sulfur metabolism and some amino acids in the response. There are two recent reports in Brassica oleracea that found overlapping results using a different approach (metabolomics).
Identification of distinctive physiological and molecular responses to salt stress among tolerant and sensitive cultivars of broccoli (Brassica oleracea var. Italica).
Chevilly S, et al.,
BMC Plant Biol. 2021 Oct 25;21(1):488. doi: 10.1186/s12870-021-03263-4.
Physiological and Molecular Characterization of the Differential Response of Broccoli (Brassica oleracea var. Italica) Cultivars Reveals Limiting Factors for Broccoli Tolerance to Drought Stress.
Chevilly S, et al.,
J Agric Food Chem. 2021 Sep 8;69(35):10394-10404. doi: 10.1021/acs.jafc.1c03421. Epub 2021 Aug 27.
Authors should include in the discussion those recent advances, which are relevant to the results presented in the report.
Author Response
Dear Reviewer:
We are very grateful to your comments for the manuscript.
According to comment, related contents have been improved. We have made revision which using the “Track Changes” function in the manuscript.
Attached please find the revised version, which we would like to submit for your kind consideration. We appreciate for Reviewers’ warm work earnestly.
And special thanks to you for your comments. I am looking forward to hearing from you.
Yours sincerely,
Huishan Liu
Corresponding author:
Name: Zongxiang Zhan
Zhongyun Piao
E-mail: zhanzxiang@syau.edu.cn
zypiao@syau.edu.cn
